# Assessment of Online Environment and Digital Footprint Functions in Higher Education Analytics

Elena Pozdeeva [1,*], Olga Shipunova [2], Nina Popova [3], Vladimir Evseev [4], Lidiya Evseeva [1], Inna Romanenko [5] and Larisa Mureyko [6]

1 Graduate School of Media Communications and Public Relations, Peter the Great Polytechnic University, 195251 St. Petersburg, Russia; evseeva_li@spbstu.ru
2 Department of Social Sciences, Peter the Great Polytechnic University, 195251 St. Petersburg, Russia; shipunova_od@spbstu.ru
3 Graduate School of Applied Linguistics, Peter the Great Polytechnic University, 195251 St. Petersburg, Russia; office@spbstu.ru
4 Institute of Physical Culture, Sports and Tourism, Peter the Great Polytechnic University, 195251 St. Petersburg, Russia; uznik_2001@mail.ru
5 Department of Philosophy, Herzen State Pedagogical University of St. Petersburg, 195251 St. Petersburg, Russia; mail@herzen.spb.ru
6 Department of History, Philosophy, Political Science, Sociology, Emperor Alexander I St. Petersburg State Transport University, 195251 St. Petersburg, Russia; dou@pgups.ru
* Correspondence: elepozd@mail.ru

**Abstract:** The article is devoted to learning analytics problems associated with the digital culture development in the university educational space and with the student activity control in the vocational training process. The empirical basis of the study was a series of surveys conducted by the Center for Sociological Research of the Peter the Great Polytechnic University in 2018–2020. To systematize the information on the traces of students' activity in the digital space, the method of constructing a personal mental map, reflecting the cognitive characteristics of the student's interactive actions in the network, was used. Because of the analysis of the mental maps, the general structure of the personal digital footprint was identified, which is significant for analytics of the student's academic history and self-assessment of his activities in professional development. In conclusion, the constructive role of digital technology in assessing and modeling the educational process is emphasized. The study of students' digital footprints on the university platforms, supplemented by the study of their activity in social networks, allows the development of educational modeling aimed at creating a more adequate set of competencies and soft skills of the graduate.

**Keywords:** digital culture; university; e-learning; digital footprint; personalized control; tools of activity in the network; mental map

## 1. Introduction

The new educational landscape of the university is defined by the development of "smart" universities. This mainstream involves all the agents involved in the educational process, with varying degrees of interest in innovative forms of relations (our article in sociology). Planning and optimization of the learning process at the university as a multi-agent system is implemented based on monitoring data from network services, e-learning clusters, and systematization of data from digital traces of students and teachers. The tools for assessing the quality of the results of professional training of a future specialist are technologies for analyzing the educational environment based on a large array of data. With the use of cloud storage technologies and the widespread introduction of information technologies, it is not so much the issue of collecting information as of its use to improve the content of general education using digital footprint monitoring and big data technologies [1–3]. The developers of commercial online courses, electronic textbooks,

or blended learning systems are already using this information most actively, adjusting the system to the student [4,5].

Descriptive analytics of the educational environment is based on an array of information exchange data between the educational process agents [6,7]. Identification of the generalized academic student group profiles, comparative data analysis from different study periods to identify students' academic history, their academic performance and the acquired competence quality allows to model future students' behavior, taking into account the identified specifics [8,9].

The development of digital footprint technology in the specialist training allows for personal control [10] and self-control in the process of building professional skills and competencies, expanding basic and non-core knowledge as the basis for learners' readiness to expand the range of competencies in the context of the intensive information environment development and the technological revolution. A digital footprint understood as an unstructured array of personal data, allows us to record the student's experience of actions in virtual space and networks, formal and informal relationships and contexts, to track the motivation of cognitive interest in the process of professional advancement and self-affirmation of a student in the information field of subject activity [11]. The functions of a digital footprint are not only control and development of digital literacy but also stimulation of critical thinking, reflection on media content in the educational process.

A systematic approach to assessing education quality using digital footprint technology is focused on the personalization prospect in the development of smart learning by the students' claims and personal inclinations concerning their professional growth [12]. Within the educational analytics framework, topical issues are related to clarifying the content of such structural elements of the electronic university environment as "digital footprint", "digital student portrait", "digital profile/passport", "route/trajectory", as well as their specificity and functionality. Otherwise, the language of constructing and explaining the strategic directions of the educational sphere development turns out to be divorced from social reality, from the agents' understanding of the essence of their activity in digital and adjacent spaces [13–15].

The purpose of the article is to study the functions of the digital footprint as an interactive agent of the educational environment of the university in students' academic history formation, as well as their self-assessment of educational activity experience and prospects in their professional sphere.

*Literature Review*

International research focuses on the institutional problems of higher education in relation to the organization of effective university management. The target audience for Learning Analytics is the university's faculty and support staff, which supports the continuity of the educational process. The constant interest for researchers and decision makers is the motivation of the faculty to introduce innovative teaching methods [16–20].

In recent years, the need to analyze students' personal experiences and self-regulating learning skills has been highlighted [21,22]. Learning Analytics includes educational technologies that emphasize student motivation and uses methods to measure the effectiveness of pedagogical approaches such as problem-based learning and project-oriented activities. Comprehensive analytics results provide a framework for personalized learning, improving academic performance and the curriculum, motivating faculty to master the forms of electronic (and blended) learning, and experiential training focused on professional communities [23,24].

Against the backdrop of the pandemic in 2020, the regime of emergency online education in higher education is emphasized. The attention of researchers is drawn to the analysis of the learning integrative structure within the communities of practice framework and the creation of communication platforms as potential artifacts of online learning [25].

Digital culture emphasizes the importance of technical systems and technologies that create a variety of the communicative world and gaming space. It participates in

the formation of sustainable social qualities of a modern personality, the adoption of behavior norms in the digital environment, including the established practices of network communication and the work with information [26,27]. Such characteristics of digital culture as variability, polymorphism, recoverability, and boundary permeability change the subject's idea not only of corporeality (the cyborg becomes a symbolic hero here) [28] (p.70), but they also influence identity by producing a "poly-image" of the subject, who, with the help of various masks (avatars) and digital traces, multiplies himself and his life in society.

The relationship between people's perception of their personality and the identity created by others in cyberspace is an important aspect for digital footprint functions study [29]. Digital culture is reflected in the formed integrated indicator of digital literacy, which includes a set of knowledge and skills necessary for the safe and effective use of computer technologies and Internet resources [30]. There are four types of digital competence: information and media competence; communicative competence.

Active digital communications entail the transformation of organizational forms of cooperation between subjects, and contribute to the development of mobility processes, complicating the social interaction space [31]. Therefore, the inclusion of digital traces and trajectories into subjects requires a sociological analysis of the new relation aspects, their socio-role content, normative and value-semantic content. M. Hoff emphasizes that digital spaces, through which young people expand their boundaries of communication and representation, develop collaborative and interactive thinking of the participants [32] (pp. 94–96).

A digital footprint is the result of a subject's activity in the digital environment, which is a unique set of actions. It can also be a passive footprint in the form of data that can be collected about users without their consent. The user forms the active footprint independently, but it can also be intentionally formed or unconscious, accompanying the publication of any important information about the subject himself. Today, a lot of information on user activity footprints is concentrated in the digital space, which makes it possible to compile models of their cognitive and psychological characteristics and use them for management purposes. It also increases the importance of the user's responsibility for the information and those actions taken by him in the digital environment [33]. In general, users retain 70% of their digital footprints, but they do the remaining 30% of their online activities inadvertently [34].

One of the learning analytics tasks is to develop scenarios of network architecture for use cases in managing a subject's digital behavior. The evaluations of Student Relationship Management system (SRMS) architecture using the Internet of Things (IoT) are based on collecting the digital footprints of higher education institutions. Because of this approach, the SRMS is refined, which consists of six main parts: (1) service station, (2) system identification, (3) API system integration, (4) internal SRM system, (5) report analytics, and 6) web server and database server. The evaluation of the system application results showed a positive trend: the usability was at a very high level, with the reviews emphasizing the effectiveness of the service support system use for students. It was concluded that this approach promoted learning and analysis of student behavior in higher education [35].

## 2. Materials and Methods

The authors use a constructivist approach as a methodological basis for analyzing the digital footprint functions in e-learning organization. This approach allows us to explore the student's attitude to the educational environment, to consider this environment from the point of view of integrating various behavioral strategies of the actors, participants in educational communications, to take into account the contexts corresponding to the young people's interests, peer culture, and the subject of study [36] (p. 34).

The objectives of the empirical research are related to identifying students' and teachers' attitudes to the e-learning formation system on the factual basis of the survey series conducted by the Center for Sociological Research of the Peter the Great Polytechnic Uni-

versity in November–December 2018. The research method is an online survey using Google forms.

The sample of the study was formed drawing on random selection of 501 respondents from the students taking different courses in the undergraduate programs. Confidence interval is 4.21%. The Center for Sociological Research has a database of students' email addresses for involving them in sociological surveys. Accordingly, the respondents to this study were students who responded to the offer to participate in the survey. The number of respondents (501 people) is enough for an intelligence and descriptive socio-logical survey, as it provides an opportunity to catch the characteristic students' attitude to the subject under study. The study also involved 39 teachers as experts who received an offer to participate in the survey on e-corporate mail. The criteria for selecting experts were at least five years of university and distance learning experience.

In the questionnaire for students, two opposite directions of evaluating online learning were presented, which allow us to identify the advantages and disadvantages of e-learning. About the merits of digital learning, it was proposed to give a subjective assessment to the following aspects of the learning process: assignment schedule flexibility, no class attendance, the ability to choose the material mastering pace, the ability to combine work with study, the absence of personal pressure. Concerning the shortcomings of digital learning, a student's assessment of learning process aspects was assumed, which concern teachers' advice quality, time limits for completing assignments, technical problems of working on the Internet, motivation to study, the pattern of performing online tasks, and the impact of online learning on eyesight.

The teacher survey assumed a personal assessment of the online learning organization quality in the following positions: difficulties in combining scientific work and the development of online courses, lack of time, lack of specialists in the implementation of online courses, weak methodological support of online learning, poor material and technical support, and inadequacy of the regulatory framework for online learning.

To analyze the students' opinions on the digital footprint functions, a study was conducted with 35 people taking part in it (SPbPU, September 2020). Two groups of students (of technical and humanitarian training) were involved in the study. They were asked to complete two tasks: (1) to draw a mental map of their activity on the Internet, (2) to give comments in the free essay form on the role of the digital footprint in learning.

Available interactive mind mapping tools (e. g. Mindmeister, WiseMapping, Drow) were used to create a personal mind map of the digital footprint. Personal mental maps were considered as the empirical material for comparing the parameters characterizing the digital footprint functions in the learning process.

## 3. Results

Overall, just over half of the students expressed a positive attitude towards the form of distance learning (55%, while 33% responded negatively). Nevertheless, over half of the respondents (54%) prefer to combine two learning formats: traditional, with personal attendance at classes and direct communication with teachers, and electronic, with a more flexible training schedule (Table 1). The results coincide with an analysis of the prospects for mixed learning in international studies, which highlight as meaningful grounds for its preference, in particular, stimulating the students' motivation to learn, the effectiveness of information assimilation in cooperation with the teacher [37,38] and more freedom in the questions and suggestions exchange [39,40].

**Table 1.** Students' Preferences of the Education Form.

| Form of Study | Positive Attitude (%) | Negative Attitude (%) |
|---|---|---|
| Distance e-learning | 55 | 33 |
| Blended learning | 54 | −0 |

Among the digital learning advantages, most of the surveyed students highlighted the flexibility of the work schedule for completing assignments (61%), the absence of the need to attend classes (60%). Less than a half of the respondents singled out the opportunity to choose the pace of mastering the material (42%), the ability to combine work and study (24%), the absence of personal pressure (21%) (Figure 1). International research also highlights the possibility of flexible, asynchronous, and remote learning as the positive side of online form [16].

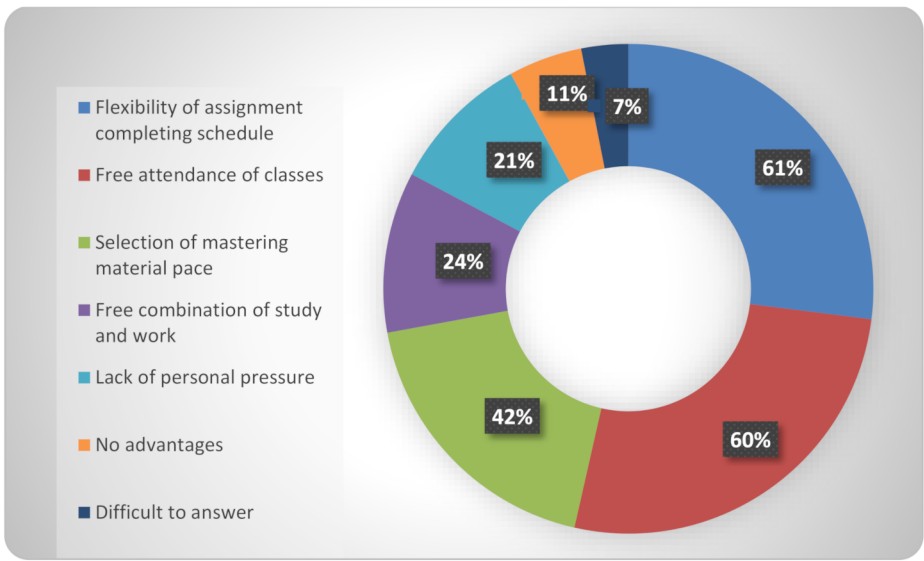

**Figure 1.** Benefits of Digital Learning as Indicated in Student Assessments.

The most pressing problems in the interactive organization of online learning, as indicated by the students, were the lack of operational communication with the teacher in case of difficulty completing assignments (58%) and meeting the deadlines for submitting answers to assignments (52%). More than 40% of the respondents noted the lack of live communication (46%), technical problems of network connection and working on the Internet (44%), as well as lack of motivation when studying at home (42%) and difficulty completing assignments in online courses (42%) as disadvantages of the e-form of education. A negative impact on eyesight was noted by a third of the respondents (31%) (Table 2).

**Table 2.** Difficulties of Digital Learning as Indicated in Student Assessments.

| Response Scale | Value (%) |
|---|---|
| Lack of prompt consultation with teachers | 58 |
| Difficulty following deadlines for completing tasks | 52 |
| Lack of communication with the teacher | 46 |
| Technical problems of connection to the network | 44 |
| Learning motivation problems | 42 |
| Difficulty completing assignments in online courses | 42 |
| Negative effect on eyesight | 31 |
| Difficult to answer | 3 |

Most of the teachers (74%) singled out difficulties associated with limited communication time as a problem of work organization in the online learning system. Over a half of the respondents (59%) indicated the problem of combining scientific work and the creation of developing online courses, 20% noted the lack of specialists in the implementation of online courses. Less than 20% of the interviewed teachers associate the problems of e-learning organization with methodological (18%), logistic (10%), and regulatory (7%) support (Figure 2).

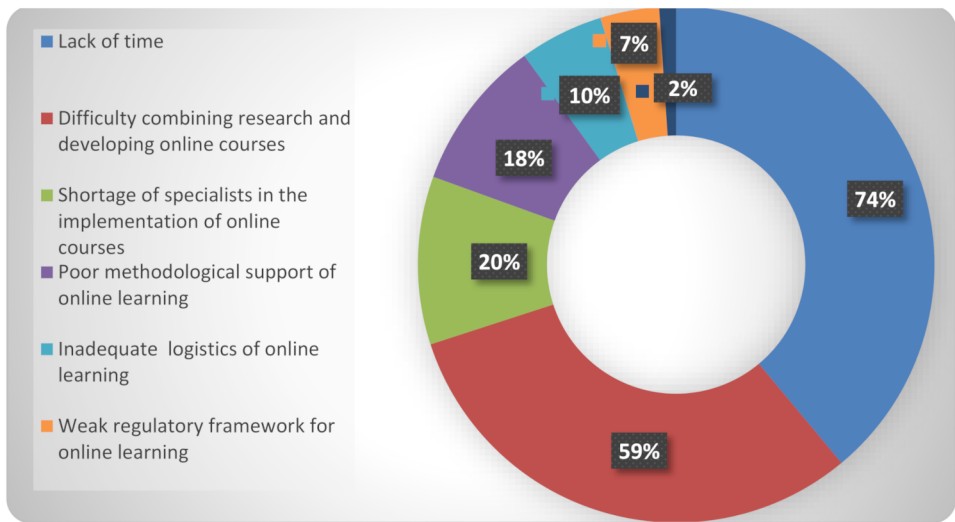

**Figure 2.** Difficulties of online learning noted by educators.

A sociological survey conducted at St. Petersburg Polytechnic University (March 2020: 73 students, 6 teachers, online survey method) was aimed at clarifying the features of the interaction between teachers and students in the digital space of the educational environment. It allowed us to form a general understanding of the difficulties and benefits of communication in the digital environment of the university. The respondents (both students and teachers) noted that they freely use digital communication channels, for both educational purposes and communication, and are aware of the risks of working in the network. At the same time, it was noted that students easily and quickly switch to different digital communication channels, and for teachers, this is more difficult. In the digital space of the university, the rules of communication on the Internet have already developed and are in effect: correspondence is carried out mainly during working hours, everyone adheres to a business style of communication. All the participants of the educational process use digital communication channels: they transfer files via e-mail, use the tools of instant messengers and social networks for urgent messages. According to the respondents, there are different preferences for methods of identification on the Internet: it is more common for students to indicate their real names and post photos, while teachers do not tend to post photos. This section may be divided by subheadings. It should provide a concise and precise description of the experimental results, their interpretation, as well as the experimental conclusions that can be drawn.

The analysis of the mental maps compiled by the students made it possible to identify the data blocks for the structure of their digital footprint in the network (Figure 3), which, from their point of view, are most important for assessing the individual activity level and the organization of the educational process in the e-learning system:

- Personal data
- Tools for displaying the presence on the Internet (social networks, mail, instant messengers, telephone, Internet services).
- Activity data: learning (in and out of the university with digital results), participation in events, various activities, creativity, sports, volunteering).
- Communication data (conducting dialogues, posts in social networks).
- Availability of official documents (in services).
- Information on consumption (all types of consumer behavior).
- Use of various services (banking).
- Moving in space (geolocation).

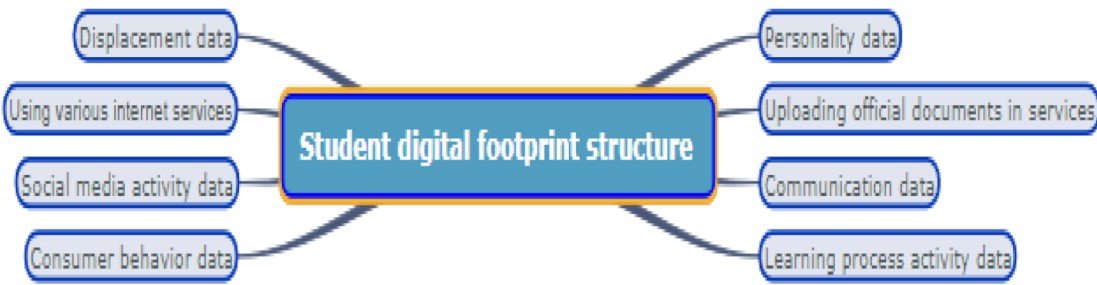

**Figure 3.** Personal digital footprint structure for learning analytics.

Mind-mapping technique in the implementation of the digital footprint technology visualizes personal presence at different stages and levels of activity in the modes of official and unofficial nature, enables us to obtain the information necessary for adjusting and planning the student performance assessment, and educational program success.

The essay analysis made it possible to identify the following semantic units that students mention when discussing the meaning and prospects for the development of a digital footprint as an element of the digital space behavior:

1. A digital footprint is presented to students in the following way: as an imprint of activity, as a set of personal data, as a student's virtual personality, as a method of digital recording and interpretation of student's actions, as a system of data storage and exchange between participants in interaction in the digital environment. Most students distinguish active and passive footprints, also noting the popularity and/or insufficient popularity of this tool, depending on the university, region, and field of study.

2. Students note the possible consequences of using a digital footprint, both positive (self-promotion, strengthening one's public role) and negative (frivolous attitude, a tool of pressure on the part of more informed people, access to personal data, tightening control, possible bullying, information leakage, service connection, targeted advertising etc.).

3. Students are well aware of the different goals of using a digital footprint for students and teachers. They note that for the university, this means the ongoing education modernization process; for a teacher, it is the analysis of the student's interests and activities. As for the students, the digital footprint helps them to search for the areas of self-development, serves as a tool for assessing the knowledge and skills acquired and contributes to their reputation and image.

4. Students see the prospects of further digital footprint use in the improvement of methods for diagnosing professional competence, in the development of new forms of digital culture, which includes a culture of interactions, new norms, and values. Students also note that a digital footprint can make an employer's search to find the right candidate more transparent.

5. As for the usefulness of a personal digital footprint for educational purposes, students note the following: the development of creative elements, project work optimization, group interaction, access to masterclasses and online events, assistance in work on final qualifying work. Moreover, they believe that digital footprint analysis is becoming necessary when an applicant enters a university.

6. As for the current problems, the students point out the inability to take into account the entire digital profile, including hobbies, volunteering and extracurricular activities. For this, a creative portfolio is proposed.

It is also worth highlighting the often-mentioned aspect associated with the employer's possible analysis of the student's digital footprint to evaluate him as a potential employee.

## 4. Discussion

The digital footprint function study develops the idea of an effective e-learning model with a cloud-based database of students' academic history and their online interactions with teachers [41].

The problem of digital manifestation of the competencies acquired by students and the types of their activities in the network, correlated with the digital footprint, was monitored in the course of experiments conducted within the framework of the "University 20.35" platform. This made it possible to clarify the structural and semantic aspects of the big data technology in educational analytics (Digital footprint: new tasks of the education system in the era of data (Electronic resource). Access mode: https://habr.com/ru/post/513616/ (accessed on 30 March 2021)). So, when processing digital footprints, it was found that unnecessary data took up a large place, which indicates the need for filters to sift through the information. The process of acquiring competencies and soft skills by students outside the classroom is almost impossible to digitize. However, it is precisely this aspect of vocational education that is emphasized by employers and is a priority today for the students themselves. At the same time, students' interest in filling the digital footprint is still weak, while this aspect presents greater interest to the functional departments of the university collecting analytics.

In the electronic information and educational environment of universities, the LMS e-learning system is actively used to manage the digital activity of participants. The main processes implementing educational and communication goals are carried out in it, with the main digital footprints being concentrated there. The accumulated experience of Russian universities in students' digital activity analysis has already made it possible to work with such parameters as the academic load regularity, students' self-organization level, their achievement deviation degree compared to the average group values. Digital footprint analysis provides a clearer picture of students' learning style, which involves assessing students' activity in online courses. Content viewing and assignment completion indicators allow you to compare students' achievements with the average group values. The currently integrated assessments allow the university to rank students into categories:

- "strong"—capable and ready to go beyond the educational program for in-depth study of disciplines and modules.
- "weak"—having academic debt, not coping with the curriculum on time.
- "special"—who have shown a high level of intellectual development and personal motivation [42].

In this study, we rely on the theory according to which ideas and thoughts act as personal constructs, since the perception and awareness of reality are associated with its interpretation [43]. A person, critically evaluating socio-cultural world objects, can design, reconstruct and deconstruct any phenomena. Experiencing certain events, the subject interprets them, structures and endows them with meanings [44,45]. In practical life, the established institutional forms act as a product of the social reality construction, which shapes the human activity horizons [46] (pp.69–86). A person's stay and activity in the virtual space, realized with the help of digital devices, is part of his constructive activity in life and learning.

Mental mapping is a popular research method that reveals the possibilities of using social construction, when a respondent acts as a constructor and represents the space, in which social activity is carried out, in the form of a map. Drawing up mental maps is accompanied by group discussion and interviews.

Mental map value is in the visualization of the consciousness constructs with the constructed space. They can serve as a starting point in social system design, in intelligence studies aimed at identifying the phenomenon contours and its primary structural representation. An important mental map advantage is in the fact that a respondent independently carries out the construction—in contrast to the mapping method when the researcher is engaged in drawing up the structure from the informant's opinion. The difference from cognitive maps actively used to study spatial problems is also significant: mental maps are

focused not only on space but also on meanings and their reflection in the structure of the phenomenon under consideration.

## 5. Conclusions

In general, the study showed a predominantly positive, interested student attitude towards the digital footprint as an educational space element. Almost all the respondents refer to the digital footprint as an already operating tool for assessing educational results. It should be noted that comprehension and reflection on such aspects of digital footprint technology application in education as responsibility, self-presentation, and image, as well as design skills are still at an early stage. This is evidenced by the fact that not all (less than half of the respondents) students pay attention to this in their mental maps and essays.

The study of students' digital footprints on university platforms, supplemented by the study of their activity in social networks, allows developing educational modeling aimed at creating a more adequate set of graduates' competencies and soft skills. Information describing the digital footprint is also valuable for the development of variable approaches to educational programs.

Students and teachers are interested in active mastering digital communication possibilities, improving the educational process using a digital footprint. However, the activity degree in creating a digital architecture of the educational space does not yet look uniform. Organizational, technical, and managerial shortcomings, awareness of the openness risks on the Internet, an unclear idea of one's "digital role" and its attributes, lack of available regulations and ethical approaches slow down the digital communication process. Along with this, an insufficiently high level of subjectivity is also noted, which is expressed in a weak initiative on the part of all the agents-participants.

The work on the formalization and semantic content of the tools for the digital educational environment development is associated with the parallel problems of creating a material and technological base, enabling us to fix, track, and assemble digital footprints and trajectories of the educational process participants, and consolidate the semantic load of indicators. However, digital culture development remains a fundamental process, which implies the subject's conscious design of his image and footprint in the digital space, barrier-free communication and the use of all the digital channel and platform possibilities. It also implies the value attitude towards one's role and active participation in the development and implementation of the regulatory framework for interaction.

**Author Contributions:** Conceptualization, E.P. and O.S.; Data curation, V.E.; Formal analysis, I.R.; Investigation, E.P.; Methodology, E.P., O.S. and L.E.; Project administration, L.M.; Validation, V.E.; Writing—original draft, N.P.; Writing—review and editing, O.S. and L.E. All authors have read and agreed to the published version of the manuscript.

**Funding:** This research received no external funding.

**Institutional Review Board Statement:** The examples given, reflecting the results of the research, are anonymous. Ethical approval was received from the Ethics Commission founded in the Institute of Humanities, Peter the Great St. Petersburg Polytechnic University, which is ruled by the code of ethics of the Russian Society of Sociologists.

**Informed Consent Statement:** Informed consent was obtained from all subjects involved in the study.

**Data Availability Statement:** Research data can be provided by the authors upon request.

**Acknowledgments:** The authors are grateful to the administration of Peter the Great Polytechnic St. Peterburg University for the opportunity to conduct research.

**Conflicts of Interest:** The authors declare no conflict of interest.

## Abbreviations

| | |
|---|---|
| SPbPU | Peter the Great St. Peterburg Polytechnic University |
| LMS | Learning Management System |
| SRMS | Student Relationship Management system |
| IoT | The Internet of Things |

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
