# Peer review of "Assessment of Online Environment and Digital Footprint Functions in Higher Education Analytics"

_education, doi:10.3390/educsci11060256_

Round 1

Reviewer 1 Report

Podziękuj autorom za przeprowadzenie tego ważnego badania ewaluacyjnego. Bardzo ciekawe są informacje o śladach aktywności uczniów w przestrzeni cyfrowej. To są mocne strony rękopisu.

Chciałbym, aby autorzy zastanowili się nad następującymi kwestiami:

  • opisać stan międzynarodowych badań dotyczących przedstawionych zagadnień. Istnieje wiele aktualnych raportów. Proszę rozszerzyć przegląd literatury
  •  nieco bardziej szczegółowo opisz metodę doboru próby uczniów. Czy był to wybór celowy czy losowy? 
  • nie powielaj danych w tabeli lub na rysunku. Moim zdaniem wystarczy jedna forma prezentacji
  • w dyskusji brakowało odniesień do raportów innych badaczy. Jak wyniki odnoszą się do literatury. Warto porównać z wynikami poprzednich badań.

Author Response

Dear Colleague!

We sincerely thank you for carefully reading our article and constructive advice on improving the text.

We have made the following adjustments and hope to have taken all your advice into account.

  1. We have supplemented the literature review with a description of the educational analytics problems in international research on issues related to the integrated assessment of the online learning environment, increased the number of sources.
  2. We correlated the results of our research with the available achievements in the international literature.
  3. We have added a description of the conditions of empirical research

The sample of the study was formed based on random selection, it made up 501 respondents - from among the students studying at different courses in the undergraduate programs. The confidence interval is 4.21%. The Center for Sociological Research has a database of students' email addresses for involving them in sociological surveys. Accordingly, the respondents to this study were students who responded to the offer to participate in the survey. The number of respondents (501 people) is enough for intelligence and descriptive sociological survey, as it provides an opportunity to catch the characteristic attitude of students to the subject under study.

  1. We took into account your remark - do not duplicate data in a table or drawing

Yours faithfully,

 Dr Elena Pozdeeva and the Team of Authors

Reviewer 2 Report

The paper is original and presents an adequate structure both in the theoretical and literature review section and in the methodological section. The results are significant for the study area, so I consider that their publication is feasible.

Author Response

(The authors gave the same response as above.)
